# FluoRNT: A robust, efficient assay for the detection of neutralising antibodies against yellow fever virus 17D

**Magdalena K. Scheck**[1], **Lisa Lehmann**[1], **Magdalena Zaucha**[1], **Paul Schwarzlmueller**[1], **Kristina Huber**[2], **Michael Pritsch**[2,3], **Giovanna Barba-Spaeth**[4], **Oliver Thorn-Seshold**[5], **Anne B. Krug**[6], **Stefan Endres**[1,7], **Simon Rothenfusser**[1,7]◎*, **Julia Thorn-Seshold**[1,5,7]◎*

1 Division of Clinical Pharmacology, University Hospital, LMU Munich, Munich, Germany, 2 Division of Infectious Diseases and Tropical Medicine, University Hospital, LMU Munich, Munich, Germany, 3 German Center for Infection Research (DZIF), Partner Site Munich, Munich, Germany, 4 Structural Virology Unit and CNRS UMR 3569, Virology Department, Institute Pasteur, Paris, France, 5 Faculty of Chemistry and Pharmacy, LMU Munich, Munich, Germany, 6 Institute for Immunology, Biomedical Center, Faculty of Medicine, LMU Munich, Planegg-Martinsried, Germany, 7 Unit Clinical Pharmacology (EKliP), Helmholtz Center for Environmental Health, Munich, Germany

◎ These authors contributed equally to this work.
* simon.rothenfusser@med.uni-muenchen.de (SR); julia.thorn-seshold@cup.lmu.de (JTS)

**Data Availability Statement:** All relevant data are within the paper and its Supporting Information files.

**Funding:** This research was supported by funds from: - Einheit für Klinische Pharmakologie (EKliP),

## Abstract

There is an urgent need for better diagnostic and analytical methods for vaccine research and infection control in virology. This has been highlighted by recently emerging viral epidemics and pandemics (Zika, SARS-CoV-2), and recurring viral outbreaks like the yellow fever outbreaks in Angola and the Democratic Republic of Congo (2016) and in Brazil (2016–2018). Current assays to determine neutralising activity against viral infections in sera are costly in time and equipment and suffer from high variability. Therefore, both basic infection research and diagnostic population screenings would benefit from improved methods to determine virus-neutralising activity in patient samples. Here we describe a robust, objective, and scalable **Fluo**rescence **R**eduction **N**eutralisation **T**est (FluoRNT) for yellow fever virus, relying on flow cytometric detection of cells infected with a fluorescent Venus reporter containing variant of the yellow fever vaccine strain 17D (YF-17D-Venus). It accurately measures neutralising antibody titres in human serum samples within as little as 24 h. Samples from 32 vaccinees immunised with YF-17D were tested for neutralising activity by both a conventional focus reduction neutralisation test (FRNT) and FluoRNT. Both types of tests proved to be equally reliable for the detection of neutralising activity, however, FluoRNT is significantly more precise and reproducible with a greater dynamic range than conventional FRNT. The FluoRNT assay protocol is substantially faster, easier to control, and cheaper in per-assay costs. FluoRNT additionally reduces handling time minimising exposure of personnel to patient samples. FluoRNT thus brings a range of desirable features that can accelerate and standardise the measurement of neutralising anti-yellow fever virus antibodies. It could be used in applications ranging from vaccine testing to large cohort studies in systems virology and vaccinology. We also anticipate the potential to translate the methodology and analysis of FluoRNT to other flaviviruses such as West Nile, Dengue and Zika or to RNA viruses more generally.

Helmholtz Center Munich, Neuherberg, Germany (https://www.helmholtz muenchen.de/forschung/ forschungseinrichtungen/klinische kooperationen/ klinische pharmakologie/forschung/index.html), to JTS, SE and SR; - LMU Center for Integrated Protein Science Munich (CIPSMwomen) to JTS - German Research Foundation (DFG): Emmy Noether grant TH2231/1-1 to OTS (https://gepris. dfg.de/gepris/projekt/400324123) - Friedrich Baur Foundation (FBS) to JTS - Ro 25257/-1 grant number 391217598 (https://gepris.dfg.de/gepris/ projekt/391217598) KR2199/10-1 to SR and ABK - SFB/TR-237-B14 Grant No. 369799452 (https:// gepris.dfg.de/gepris/projekt/404450088) to SR and ABK Furthermore, we would also like to acknowledge support from the international doctoral program "iTarget: Immunotargeting of cancer" funded by the Elite Network of Bavaria to MKS, MZ, LL, and SE. MKS gratefully acknowledges a PhD fellowship from the Max-Weber Foundation (https://www.elitenetzwerk. bayern.de/start/foerderangebote/max-weber-programm) and support from the LMU FöFoLe and Lehre@LMU programmes. MP was supported by a Metiphys fellowship of the Medical Faculty of the LMU Munich. The funders had no role in study design, data collection and analysis, decision to publish, or preparation of the manuscript. JTS was receiving salary from the Einheit für Klinische Pharmakologie (EKliP), Helmholtz Center Munich at the time the work was performed. OTS is receiving salary from German Research Foundation (DFG).

**Competing interests:** The authors have declared that no competing interests exist.

## Introduction

In light of recently emerging viral epidemics and pandemics (Zika, SARS-CoV-2), it has become urgent to develop quick, reliable and scalable assays to determine neutralising antibodies indicative of protection against infection. These assays should be suitable for testing large numbers of patient samples, using automated and objective evaluation methods, which can be applied for diagnostics [1]. Such assays are important for a range of applications ranging from fundamental systems biology research on model viruses to practical testing of protection status after vaccination. The latter application is increasingly relevant: limits to vaccine production and supply have already forced the temporary use of diluted vaccine stocks in emergency mass immunisation campaigns against yellow fever starting 2016–17 [2]; and rapid scalable testing to determine successfully protected subpopulations may become a valuable aspect of epidemic control.

Yellow fever virus (YFV) is an enveloped positive-sense single-stranded RNA virus that has historically been responsible for large disease outbreaks with high morbidity and mortality across Africa and South America [3]. YFV was the subject of high-intensity research from the 1800s onwards, and in 1901 it became the first human virus ever to be isolated [4]. Max Theiler generated the live-attenuated vaccine strain YF-17D in the 1930ies [5], which confers lifelong YFV immunity in more than 95% of vaccinees and remains one of the most effective vaccines ever developed [6]. As the prototypical flavivirus and the first flavivirus against which a highly potent vaccine was developed, YFV has for a long time been a model for systematic patient studies of viral infection, innate and adaptive antiviral immune response, and immunisation [7, 8]. Such research relies on quantitative assays that robustly determine kinetics and magnitude of protective antibody response after exposure or vaccination. Quantitative assays have performance requirements far above those for qualitative diagnostic assays to determine protection status after vaccination.

A protective titre against YFV infection is most commonly measured by neutralisation tests, where the presence of neutralising antibodies results in the reduction of an infection equivalent within the assay. The serum dilution at which patient serum neutralises a set percentage of viral infection is called the neutralising antibody titre. For YFV WHO guidelines require patient sera to show at least 80% neutralisation at 1/10 serum dilution to be considered protective, i.e. the $ED_{80}$ (**e**ffective **d**ilution at **80%** neutralisation, also $EC_{80}$ or $NT_{80}$) titre should be below 1/10 [9, 10]. For the two most commonly used reduction neutralisation tests (RNTs), i.e. the focus reduction neutralisation test (FRNT) and plaque reduction neutralisation test (PRNT) for YFV, cells are treated with a fixed amount of infectious viral particles inoculated with serially diluted patient serum. After incubation, infection equivalents are quantified, referenced to an assay control and analysed as a function of serum dilution. The resulting dose-response curve allows the reading of the $ED_{80}$ neutralising antibody titre in serum (or, similarly, e.g. $ED_{50}$). **Fig 1A** shows a schematic representation of the three RNTs that are discussed in this work. The assay formats and timelines for YFV-17D PRNT and FRNT depicted here are based on previously published standard protocols adapted by our lab [11–13] and generally depend on the cell lines and virus stocks used for the respective assays.

The traditional **P**laque RNT (PRNT) counts large plaques that develop in a monolayer of cells after multiple rounds of infection by any cytopathic virus strain, as the viral infection equivalent. YFV PRNT employs a cell monolayer in 6- to 24-well format exposed to virus and diluted test serum, coated with a viscous medium overlay such as carboxymethylcellulose (CMC). The non-neutralised fraction of virus infects cells; and the viscous coating eliminates convective viral transport but allows radiative cytopathic spreading during subsequent rounds of infection. Depending on the virus strain, on the used cell line (e.g. BHK or Vero cells) and

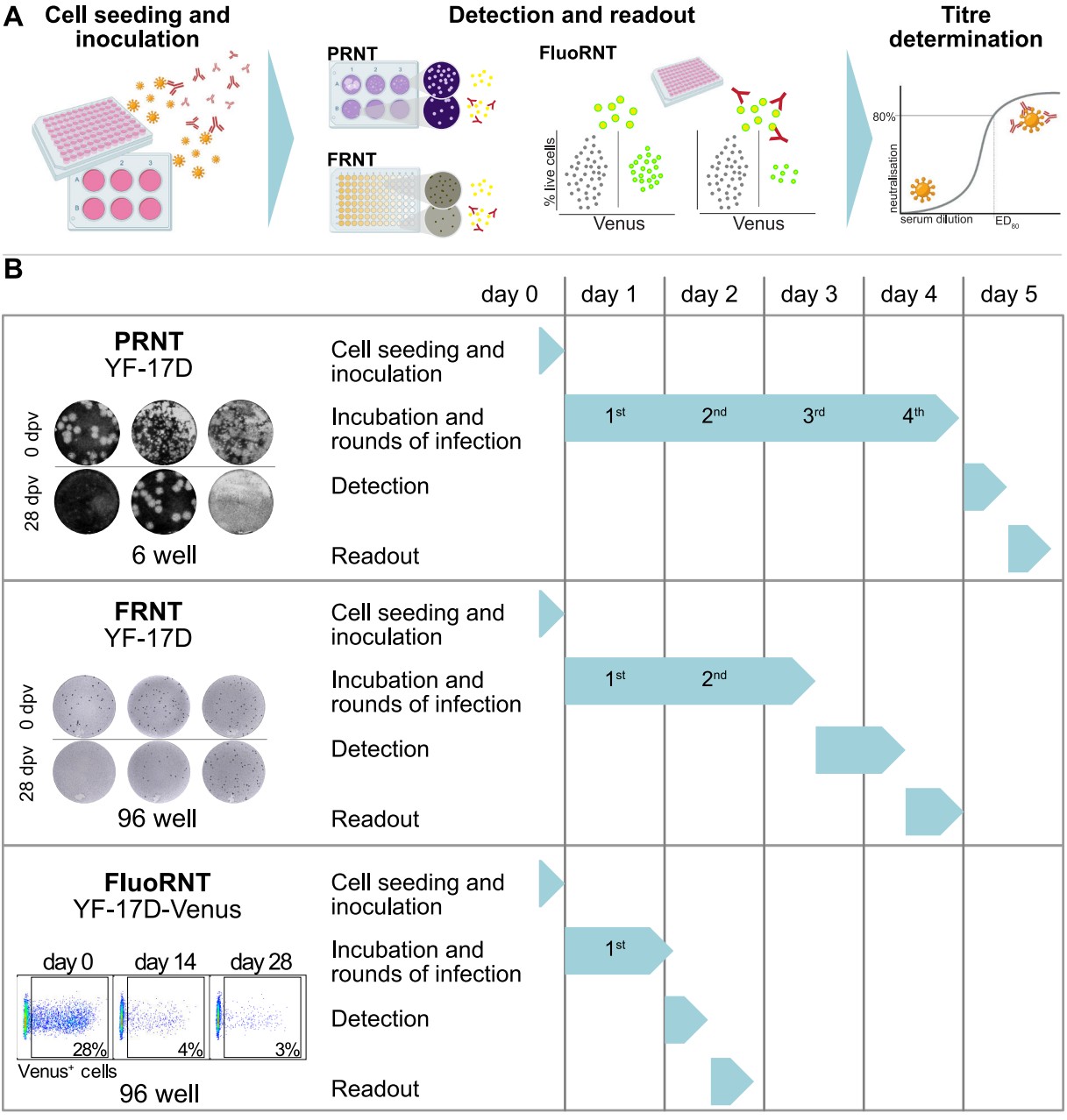

**Fig 1. Comparison of scientific and practical features of the reduction neutralisation assays FluoRNT, PRNT and FRNT.** (A) PRNT, FRNT, and FluoRNT are differentiated in their readout modes. (B) PRNT, FRNT, and FluoRNT are also differentiated in assay workflows and time requirements. FluoRNT has several practical advantages, including (i) avoiding multiple rounds of infection (2nd– 4th rounds of infection in FRNT and PRNT) that otherwise introduce and enhance assay variability; (ii) avoiding CMC addition and removal (PRNT and FRNT), staining (PRNT) or immunostaining (FRNT), and manual counting (PRNT) or image post-processing (FRNT): which are all manual steps that increase the probability to introduce handling errors. Additionally, FluoRNT (iii) is faster overall and has reduced hands-on assay time; (iv) is adaptable for high-throughput use; and (v) its assay progress can be followed noninvasively during run time.

on the MOI used, the assay is run for 4–5 days so that multiple rounds of infection give visually detectable plaques in the monolayer, that encircle originally infected cells [14–16]. PRNT read-out is then obtained by aspirating the viscous coating and staining live cells to enable counting

dead plaques. Determining the percentage of reduction of cytopathic plaque formation for a range of serum dilutions allows the fitting of a dose-response curve to determine the $ED_{80}$. The reagents and equipment used in PRNT are cheap; but the assay has major drawbacks: it requires much hands-on time, is technically highly variable, and the overall assay run time is long. These and other factors make PRNT prone to numerous sources of variation and error in assay setup and analysis, while additionally being poorly scalable.

The **F**ocus RNT (FRNT) is a 96-well format assay counting much smaller infected cell *foci* after fewer rounds of infection, visualised by immunostaining; while being similar in setup its throughput can be higher than that of PRNT, and FRNT can also be applied to any cytopathic and non-cytopathic virus for which antibodies exist [12, 17]. Assay setup for the YFV FRNT is similar to that of PRNT, including the need for viscous overlay, but multicellular foci are formed typically after 2–3 days of incubation. After methylcellulose removal and washing, immunostaining has to be performed (primary: anti-virus e.g. 4G2 clone, secondary: typically, enzyme-conjugated for chromogenic staining) [18]. The converse signal-to-background pattern of absorbent foci against an unstained cell layer should enable automated focus counting via a scanning EliSpot-type plate reader. However, in practice, high background from chromogenic staining and low signal depending on the primary antibody makes manual checking of staining results and of image post-processing obligatory in our hands. The cost of the required antibodies also limits its practicality, and the overall savings in time associated with reducing the number of rounds of infection are offset by extra washing and staining steps.

The motivation for the present study has been to develop a robust, quantitative, and scalable assay that avoids the disadvantages affecting plaque and focus reduction neutralisation tests, and better fulfils current higher-throughput needs in basic and applied virology research specifically for YFV, as well as more broadly for other viruses.

Our **Fluo**rescence RNT (FluoRNT) uses a reporter variant of the YF-17D vaccine virus to allow stain-free detection of reporter expression in infected cells after the first round of virus infection, in a 96-well format. There is no need for application and then removal of a viscous overlay; and infected cells can be reliably quantified by flow cytometry based on the yellow fluorescent reporter Venus, without additional staining or immunostaining (**Fig 1**).

In this study we focus on evaluating assay performance based on three criteria: (1) reliability, (2) practicability, and (3) data quality. We expect many of the findings to apply not only to neutralisation tests for YFV, but also to neutralisation tests for other viruses.

## Methods

### Human samples

Human sera before and after vaccination with the YFV Vaccine Stamaril® (Sanofi) were derived from a YF-17D vaccination study, approved by the responsible institutional review board of the Medical Faculty, LMU Munich; (IRB #86–16). In this study, blood was taken from healthy adults directly before the vaccination (d0) as an individual reference and on day 7, 14 and 28 post vaccination (dpv) to determine the titre of neutralising antibodies. Serum was collected in S-Monovettes (Z-Gel; Sarstedt, Nuembrecht, Germany) and separated from whole blood by centrifugation at $2500 \times g$ for 10 min. Samples were frozen and kept at -80˚C until use. To probe the reliability of the FluoRNT at later time points after the vaccination blood samples were taken from 15 healthy volunteers with a history of being vaccinated with Stamaril® between 4 month and up to 19 years ago (approved by the IRB of the Medical Faculty, LMU Munich, 24062006GP).

## Cells

Vero cells (ATCC CCL-81) and BHK-21 cells (ATCC CCL-10) were cultured in Dulbecco's modified Eagle medium (DMEM) containing 10% foetal calf serum (FCS), 1% L-Glutamine 100 IU/mL Penicillin and 100 µg/mL Streptomycin under standard conditions (37˚C, 5% $CO_2$, >90% humidity). All cell lines were regularly tested and only used if mycoplasma negative.

## YF-17D-Venus production

The YF-17D fluorescent reporter variant YF-17D-Venus plasmid was a kind gift from Charles M. Rice and Margaret MacDonald (The Rockefeller University, New York, USA). *pYF17D-5'C25Venus2AUbi* encodes a Venus fluorescent protein in frame after the first 25 amino acids of the YFV capsid gene, followed by a foot-and- mouth disease virus (FMDV) 2A peptide to mediate protein cleavage and a ubiquitin (Ubi) monomer followed by the complete YFV poly-protein sequence [19]. YF-17D-Venus genome was cloned into a *pACNR* backbone using Gibson Assembly Master Mix (New England Biolabs, Ipswich, USA).

For YF-17D-Venus production the construct (*5'C25Venus2AUbi*) in *pACNR* was linearized with AflII and *in vitro* transcribed into sense RNA using the SP6 promotor and the mMessage mMachine Kit (Ambion by life technologies, Carlsbad, USA) according to the manufacturer's protocol. BHK-21 cells were transfected with YFV-Venus RNA by electroporation. Briefly, $6 \times 10^6$ cells in 400 µL ice-cold PBS were mixed with 2 µg *in vitro* transcribed RNA and transferred into a 0.4 cm gap electroporation cuvette. Electroporation was performed immediately with 1200 V for 100 µs with 2 pulses in 5 s intervals in a Gene Pulser Xcell Electroporation System (Bio-Rad, Hercules, USA). After recovery for 10 min at room temperature cells were added dropwise into 10 cm petri dishes filled with pre-warmed DMEM containing Penicillin/Streptomycin and L-Glutamine. The success of the electroporation was verified with a Leica TCS SP5 confocal system (Leica Microsystems, Wetzlar, Germany). Yellow fluorescent cells could be seen 3–4 days after electroporation, and supernatant was harvested upon onset of cytopathic effect, after 6–7 days. The supernatant was clarified from cellular debris by centrifugation at $400 \times g$ for 5 min and further purified using a 0.45 µm filter. Aliquots of this primary virus culture were stored at -80˚C.

## Virus stock production

BHK-21 cells were grown in DMEM containing 10% FCS, 1% L-glutamine, 100 U/mL Penicillin, and 100 µg/mL Streptomycin and infected with YF-17D (derived from a Stamaril® vaccine dose) or YF-17D-Venus (see above) at an MOI of 0.1 at a confluency of 80%. Supernatants were collected post-infection when a cytopathic effect was clearly visible (after 48 h for YF-17D and after 72–96 h for YF-17D-Venus). The supernatant was clarified to remove cellular debris by centrifugation at $400 \times g$ for 5 min and further purified using a 0.45 µm filter. Aliquots were then stored at -80˚C.

For some experiments an additional purification of the virus stock was performed, adapted from previously described protocols [20]. Briefly, cell culture supernatants from virus producing cells were centrifuged at $2200 \times g$ at 4˚C for 15 min. Polyethylene glycol (PEG 8000) was gradually added to the clarified supernatant on ice (final concentration 7% w/v) and rotated at 4˚C with low speed for at least 3 h. Supernatant containing PEG was centrifuged in a Sorvall 75006445 Rotor at max speed ($\sim$3800 × g) at 4˚C for 80 min, the pellet resuspended in cold TNE buffer (20 mM Tris-HCl pH 8, 150 mM NaCl, 2 mM EDTA) and layered on top of a double sucrose cushion of 30% sucrose on top of 60% sucrose (w/w in TNE) in 5 mL open-top thinwall ultra-clear tube (Ref 344057, Beckman Coulter, Brea). Ultracentrifugation was performed at 160,000 × g at 4˚C for 2 h in an MLS 50 rotor (Beckman) and resulted in a clear

virus band between the two sucrose cushions. Pellet was resuspended in cold TNE buffer and stored at -80˚C until use.

## FRNT

The focus reduction neutralisation tests for YFV-17D and YFV-17D-Venus were performed similar to previous descriptions [11, 12, 17, 18].

**On the first day**, Vero cells were seeded in a 96-well plate with 25,000 cells/well and incubated over night at 37˚C.

Prior to infection **on the second day,** human sera were thawed, diluted 1:5 in DMEM without supplements and heat inactivated at 56˚C for 30 min. 60 μL of 1:3 serial dilutions of the sera in DMEM supplemented with 0.02% BSA were mixed with an equal volume of YF-17D or YF-17D-Venus diluted in DMEM with 0.02% BSA containing a predetermined amount of infectious units (approximately 100 focus forming units; FFU; or approximately 220 PFU of our YF-17D-Venus stock) per well. Human sera were assayed in two replicates in dilutions reaching from serum dilution $\log_{10}(ED) = -1.0$ to -5.8 for the YF-17D FRNT and from -1.0 to -4.8 for the YF-17D-Venus FRNT. No-serum controls were included using DMEM with 0.02% BSA instead of serum. The mixture was kept at 37˚C for 1 h. Afterwards, 50 μL of the inoculum was used to replace the medium of the Vero cells. To block convective viral distribution, after 1 h, 150 μL of pre-warmed methylcellulose (MCS) medium was layered on top of the cells. Briefly, MCS medium contained 1.5% MCS powder (Sigma, M0512-10G, 4000 cP viscosity) in MEM (supplier), 10% FCS, penicillin (100 U/mL), streptomycin (100 μg/mL), L-glutamine (4.5 g/L), and sodium hydrogen carbonate (3.7 g/L) to reach a pH of 7.4 in Millipore water. After infection the cells were incubated at 37˚C for 48 h.

**On the fourth day,** the methylcellulose cover was removed, the cells were fixed in 5% PFA in PBS for 1 h at room temperature, permeabilized for 5 min and blocked for 10 min using 0.5% Triton X-100 and 50 mM Ammonium chloride solution in PBS, respectively. Washing with PBS was included after each step. Foci for the YF-17D FRNT were stained with 1 μg/mL Flavivirus anti-E 4G2 antibody (CG 0042, mouse monoclonal, Clonegene, Atlanta, USA; Antibody Registry AB 2722738) diluted in 0.05% Tween in PBS (15 ± 1 h) at 4˚C overnight, followed by an anti-mouse horseradish peroxidase (HRP)-conjugated secondary antibody (7076S Cell Signalling, Danvers, USA; Antibody Registry AB 330924) diluted 1:100 in 0.05% Tween in PBS for 3 h at room temperature **on the fifth day**. We visualised the foci with a chromogenic reaction using 3,3'- diaminobenzidine (DAB) (D5905, Sigma-Aldrich, St. Louis, USA) in TBS buffer containing approximately 0.3% NiCl solution for 25 min at room temperature.

For the YF-17D-Venus FRNT foci were stained with a rabbit anti-GFP antibody (ab6556, Abcam, Cambridge, UK) diluted 1:2,500 in 0.05% Tween in PBS over night at 4˚C and an anti-rabbit horseradish peroxidase (HRP)-conjugated secondary antibody (1706515, rabbit polyclonal, BioRad, Hercules, California; Antibody Registry AB 11125142) diluted 1:250 in 0.05% Tween in PBS for 1 h at room temperature. Foci were again visualised using 3,3'- diaminobenzidine for 10–15 min at room temperature.

Foci were counted with an EliSpot Reader ELR04 SR (AID, Autoimmun Diagnostika GmbH, Straßberg, Germany). Virus neutralisation was determined as

$$virus\ neutralisation = 1 - \frac{foci\ in\ well\ on\ day\ x\ after\ vaccination}{foci\ in\ reference\ well} \tag{1}$$

for each dilution. As stated in the figure legends we used either individual pre-vaccination value 0 dpv or the mean of the run-average no-serum controls (NSC) as the reference controls.

Once virus neutralisation results reached zero, they were constrained to zero in all following dilutions. With the results of each dilution, curves with a bottom constraint greater than 0 and a top constraint less than 100 were fitted by nonlinear regression analysis (Four parameter, variable slope dose-response model: *Find ECanything*; F = 80*)* using Prism 8 (GraphPad, La Jolla, CA, USA) software. Titres were interpolated at 80% virus neutralisation (ED$_{80}$).

## FluoRNT

For the fluorescence reduction neutralisation test (FluoRNT) the assay setup is as follows:

**On the first day,** Vero cells were seeded in a 96-well plate with 25,000 cells/well and incubated over night at 37˚C.

Prior to infection **on the second day**, human sera were thawed, diluted 1:5 in DMEM without supplements and heat inactivated at 56˚C for 30 min. 60 μL of 1:3 serial dilutions of the sera in DMEM supplemented with 0.02% BSA were mixed with an equal volume of YF-17D-Venus containing a predetermined amount of infectious units to induce an infection rate of 25% ± 5% in the virus quantification, as previously reported [16]. The amount of virus used in the FluoRNT is 22-fold more concentrated than the YF-17D-Venus stock used in FRNT and corresponds to approximately 5000 PFU/well. Human sera were assayed in two replicates in dilutions reaching from serum dilution $\log_{10}(ED)$ = -1.0 to -3.9. The mixture was incubated for 1 h at 37˚C. No-serum controls as reference and negative controls were included using only DMEM with 0.02% BSA instead of serum. The medium of Vero cells was removed and replaced by 50 μL of the inoculum.

**After an incubation for 24 h** at 37˚C the medium was removed, cells were washed, trypsinised and transferred into a 96 U-well plate. For flow cytometry analysis, cells were stained for 30 min at 4˚C with a viability dye using 50 μL of APC-Cy7 Fixable Viability Dye (eBioscience™ Fixable Viability Dye eFluor™ 780, Invitrogen, Carlsbad, USA) diluted 1:5,000 in FACS buffer (PBS with 2% FCS, 2 mM EDTA and 0.1% NaN$_3$). Cells were then washed and the virus was inactivated by 4% PFA in PBS. After incubating for 25 min at room temperature cells were again washed and analysed by FACS by flow cytometry (LSRFortessa Flow Cytometer, BD, San Jose, USA) or stored at 4˚C in the dark for analysis on the next day. The infection rate per well was determined as the percentage of Venus-positive cells of all living single cells in one well. The gating strategy is shown in **S5 Fig**.

Virus neutralisation was determined according to the following formula for each dilution:

$$virus\ neutralisation = 1 - \frac{\%\ of\ infected\ cells\ in\ well\ on\ day\ x\ after\ vaccination}{\%\ of\ infected\ cells\ in\ reference\ well} \tag{2}$$

As stated in the figure legends we used either individual pre-vaccination value from 0 dpv or the mean of no-serum controls (NSC) as the reference values. Once virus neutralisation results reached below zero, they were set as zero as well as the following lower dilutions results regardless whether they reached again above zero or not. Curves with bottom constraint greater than 0 and top constraint less than 100 were fitted by nonlinear regression analysis (Four parameter, variable slope dose-response model: *Find ECanything*; F = 80) using Prism 8 Software (GraphPad, La Jolla, CA, USA) software. Titres were interpolated at 80% virus neutralisation (ED$_{80}$).

## Statistics

Where shown, the statistical tests used are noted in the figure legends. By default, tests were performed as non-parametric tests, not assuming a normal distribution of the data. P-values smaller than 0.05 were considered significant.

## Data analysis

Calculation of 80% **neutralising titres** ($ED_{80}$) for both assays is described in the respective paragraphs on assay protocol. All titres are represented as the $\log_{10}(ED_{80})$, starting from the initial 1/10 serum dilution ($\log_{10}(ED_{80})$ = -1), to 1/100 ($\log_{10}(ED_{80})$ = -2), 1/1000 ($\log_{10}(ED_{80})$ = -3) and so on.

The **assay dynamic range (ADR)** provides the range in which the assays are free from technical scatter and therefore reliable differentiate data points, by taking the technical scatter of bottom and top plateau values into account. The bottom scatter is calculated with primary data values (i.e. foci for the FRNT or Venus$^+$ cells for the FluoRNT) normalised to the run's average maximum infection values (no-serum controls, NSC). For the top scatter the technical replicates for the maximum protection values (i.e. serum in $\log_{10}(ED)$ = -1) are normalised to the run-average NSC, then the greater value is divided by the smaller one. Both scatters ideally show a tight distribution around 1. For visual representation in **Fig 3D** both scatters were transformed to the same scale. To give the ADR, the separation between the 10[th] percentile of the bottom scatter and 90[th] percentile of the top scatter (marked with "d" in **Fig 3D**) is divided by the sum of both 10-90[th] percentile widths ("s$_1$" and "s$_2$" in **Fig 3D**). As a numerical example, a near-ideal assay that generates high-quality ED fits could have bottom scatter values with 10[th]/90[th] percentiles at 95% / 110% infectious equivalents normalised to run-average NSC and top scatter values with 10[th]/90[th] percentiles at 0% / 5% infectious equivalents referenced to run-average NSC, giving a good ADR of (95–5) / ((110–95) + (5–0)) = 4.5. A highly scattered assay that is poorly suitable for $ED_{80}$ fitting might have the corresponding 10[th]/90[th] percentile plateau values at 85% / 125% and 0% / 20%, giving a poor ADR of 1.1.

To analyse assay **precision,** we tested technical replicates in the range of 55–90% neutralisation for their reproducibility. The area between 55–90% neutralisation is the most relevant region of the dose response fit to define the patient's $ED_{80}$ and therefore the titre required as correlate for protection. We therefore paid extra attention to the technical reproducibility of data points in this region. The absolute neutralisation difference $\partial$ between technical replicates in this area of interest was calculated and weighted as a precision parameter $(1-\partial)^2$.

## Software

FlowJo Software (FlowJo LLC, BD, San Jose, USA) was used for flow cytometry analysis. Confocal microscopy pictures and live image videos were analysed with FIJI [21] software. Parts of **Fig 1** and the graphical abstract were created with BioRender.com.

## Results

### FluoRNT detects protective titres with equal reliability as the benchmark FRNT assay

To test whether the first-round of infection assay FluoRNT is suitable as an alternative to multiple-round of infection assays for determining YFV protection status according to WHO criteria, we used it to measure neutralising activity after YF-17D vaccination in a cohort of 32 YFV-naive volunteers. Neutralising antibodies are formed following peak viremia and usually do not occur before 8 days post vaccination (8 dpv) [22]. According to the WHO up to 80% of vaccinees demonstrate a neutralising antibody titre by 10 dpv ($ED_{80} \leq$ 1/10 serum dilution), and most studies show that > 99% of vaccinees develop a protective titre by 28 dpv [23]. We analysed serum samples pre-vaccination and at 7 and 28 dpv, and compared the resulting antibody titres determined by FluoRNT to those measured by the benchmark assay FRNT. As expected, both assays showed no neutralising titres in the 7 dpv samples, while all volunteers

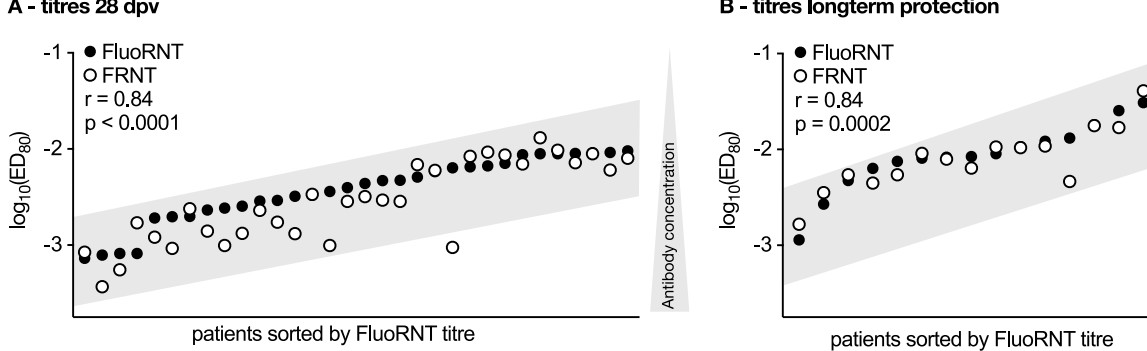

**Fig 2. FluoRNT reliably detects neutralising activity.** (A) FRNT and FluoRNT yield strongly correlating titres within the same order of magnitude and with similar distributions at 28 dpv (n = 32). (B) FluoRNT has equal reliability to FRNT in determining titres from long-term vaccinees (n = 15). Spearman r. All titres are referenced to NSC. The grey box illustrates that datasets from both assays follow the same general trend.

had protective titres at 28 dpv. **Fig 2A** shows titres measured by both assays, showing a high correlation (r = 0.84, p < 0.0001) in high as well as low titre samples indicating that FluoRNT can detect protection status with equal reliability as FRNT. To probe the applicability of both assays we analysed serum samples from vaccinees 4 month to 19 years after they had been vaccinated with YF-17D (n = 15). Titres of the long-term vaccinees were on average lower than those of the primary study group at 28 dpv. FluoRNT gave a titre median of $\log_{10}(ED_{80})$ = -2.08 for the long-term vaccinees and $\log_{10}(ED_{80})$ = -2.38 for the primary study group. FRNT gave a titre median of $\log_{10}(ED_{80})$ = -2.10 in the long-term study group and -2.54 for the primary study group. Both assays detected protective titres in all of the samples and the high correlation coefficient (r = 0.84) between the results determined by the two assays was preserved (**Fig 2B**).

While the absence or presence of protection is a useful qualitative result, quantitative assay development instead requires determining the actual antibody titre reliably and comparably, with emphasis on assay robustness, data quality, and assay practicability. We first assessed and compared assay practicability between FRNT and FluoRNT.

## FluoRNT offers significantly greater assay practicability than FRNT and PRNT

In addition to assay robustness and data quality, it is important for assay selection to evaluate practicability. **Fig 1** shows the time line for PRNT, FRNT and FluoRNT. The following features that differentiate the ease of use and implementation of these assays will be considered in more detail in the discussion section. Of note, only FRNT and FluoRNT were evaluated and compared directly experimentally; the comparison to PRNT is from previous experience and literature.

1. FluoRNT delivers results distinctly faster than FRNT and PRNT (fewer (handling) steps, shorter run time).

2. FluoRNT readout by flow cytometry is <u>objective and quantitative with the potential for automation</u>

3. FluoRNT cuts assay costs by 84% compared to FRNT (**S1 Table**) and scales up with the most affordable lab consumables

4. FluoRNT can be monitored non-invasively during assay run time

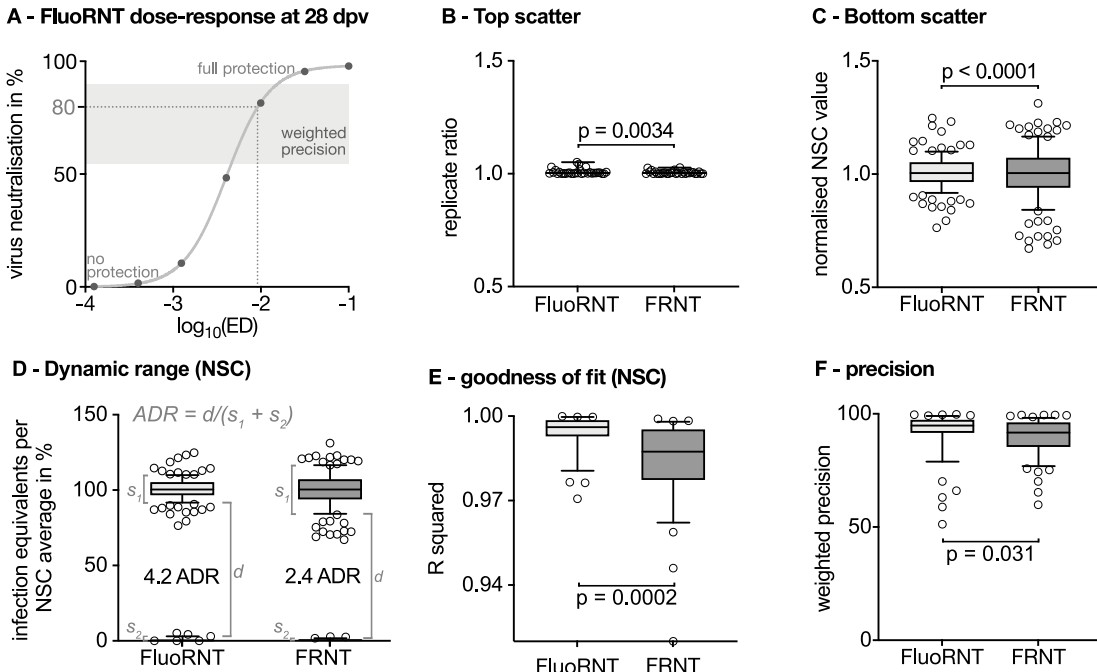

**Fig 3. FluoRNT plateau data quality and reliability is superior to that of FRNT.** If not described otherwise, all plots are box (25%/75%) and whiskers (10%/90%) plots with data outside the whiskers shown as individual data points. (A) Representative FluoRNT dose-response curve for a patient serum at 28 dpv. The "top" represents values of full protection whereas the "bottom" values of the curve show no protection against YF-17D. The grey area bracketing the $ED_{80}$ is used to calculate the weighted precision of technical replicates (see panel F). (B) Scatter of top plateau: Replicates of infectious equivalents (number of foci or percentage of Venus[+] cells) on 28 dpv at $\log_{10}(ED) = -1$ referenced to run-average NSC values, then the greater value is divided by the smaller one (F-test; n = 31). Box and whiskers show minimum and maximum of all data points. (C) Scatter of bottom plateau: infectious equivalents for no-serum controls (NSC) normalised to the run-average NSC (F-test; n = 128;). (D) Assessment of the assay dynamic range (ADR) combining the top and the bottom plateau scatter. Box plot that scatters around 100% displays the bottom scatter as in C (i.e. NSC; n = 128) and box plot that scatters around 0% represents the top plateau scatter as in B brought to the same scale (i.e. replicate ratio of 28 dpv serum at serum dilution $\log_{10}(ED) = -1$; n = 31). The dynamic range is calculated as range between the 10[th] and 90[th] percentile of the upper and lower box (see "d"), respectively and referenced to the distribution widths of both ("s_1" and "s_2"). The ADR of the FRNT assay is 2.4 while that of FluoRNT is 4.2. All values are referenced to run-average NSC. Panels B-D show technical replicates of study participants of the main cohort. (E) Goodness of fit parameter $R^2$ for data fit to a 4-parameter sigmoidal dose-response model (n = 32). Median $R^2$ for FluoRNT is 0.996 and 0.988 for FRNT. Mann Whitney test. (F) Precision of FluoRNT is significantly higher than that of FRNT (58 data points considered for FluoRNT with median of 94.91%, 66 for FRNT with a median of 91.74%). Mann-Whitney test. NSC served as reference for calculation in all panels.

In addition to showing the improved practicability of FluoRNT in its assay setup, we also tested for improvements in data quality delivered by FluoRNT, analysing reproducibility, assay dynamic range and precision compared to the benchmark FRNT.

## Assay precision and data quality

**FluoRNT yields superior plateau data quality for dose-response analysis.** To quantify ED titres reliably, data values must not only be precise (reproducible and free from random scatter) but must also give a high-quality dose-response fit (**Fig 3A**). This is prototypically a 4-parameter sigmoidal fit, with top and bottom plateaus at high and low serum concentrations respectively, with a characteristic midpoint ($ED_{50}$) and gradient at this midpoint (Hill slope). The top plateau data points correspond to complete virus neutralisation and therefore maximum protection, i.e. 100% virus neutralisation as defined by the absence of equivalents of infection (number of Foci for FRNT and percentage of Venus-positive cells in FluoRNT) in

the presence of a usually less diluted serum sample (e.g. 1/10 serum dilution). The bottom pla-teau data points correspond to no protection (i.e. 0% virus neutralisation) and are represented by more diluted serum dilutions that approach the number of equivalents of infection of no-serum controls (NSC, no serum or antibodies present) or pre-vaccination serum samples. In our calculations, equivalents of infection of tested serum are referenced to the average NSC run in the same experiment. Values referenced to pre-vaccination samples (0 dpv) can be found in the supplementary information.

In an ideal assay the top values that show maximum protection (in our case meaning the infectious equivalents on 28 dpv in the highest concentration, i.e. $\log_{10}(ED) = -1.0$ (= 1/10 dilu-tion) would give identical technical replicate values. To assess the scatter of the top values we normalise the two replicates to the run-average NSC value, then divide the greater value by the smaller. The FluoRNT and the FRNT both show a tight distribution around 1 indicating that the top scatter is nearly ideal for both assays and allows a confident determination of the top plateaus of the dose response curve (**Fig 3B**).

To assess the quality of the bottom plateaus, we analysed the NSC values (maximum possi-ble infection equivalents or minimum protection per assay) by plotting the ratios of raw data NSC values to the average NSC counts of infection in all assays that were run simultaneously (**Fig 3C**). In a high-quality assay, NSC values should show a tight distribution with little devia-tion arising from technical variability inherent to the assay. The normalised FluoRNT NSCs give a tight distribution, highlighting the reproducibility of the bottom plateaus they define (90% of data points cluster between 0.92 and 1.1 of the normalised NSC average). In contrast, the FRNT NSC values consistently spread over a wider range (90% of data points lie between 0.84 and 1.17), which could make its bottom plateau definition, and thus the subsequent dose-response fitting, problematic. This difference in the data distribution between the NSC values in FluoRNT versus FRNT stayed consistent when we tested different virus batches including batches that were purified or not by a sucrose gradient. (**S2 Fig**). Therefore, FluoRNT shows a consistently more robust bottom plateau than the FRNT.

**FluoRNT yields a greater assay dynamic range for determining ED titre values.** To study antibody response quantitatively, ED titres must be interpolated from dose-response curves, which are fitted to neutralisation values intermediate between the top and bottom pla-teaus. An ideal assay has fully reproducible full- and no-protection plateaus so that any inter-mediate data values can be reliably used for fitting the intermediate region of the dose-response curve. Fitting to intermediate values becomes increasingly imprecise and inaccurate as data scatter increases. As an extreme example, if top and bottom plateau data values are so scattered that they can be found over the full range of protection values, then confident or meaningful fitting to intermediate data values is not possible, regardless of their precision.

We therefore expected that FluoRNT would allow better and more reliable dose-response titre fits than FRNT. To quantify the quality of data fitting for ED determination that is possi-ble in each assay, we defined an **assay dynamic range (ADR).** Briefly, the dimensionless ADR is the ratio of the area in which data points suitable for ED fitting can be reliably differentiated from the technical scatter (i.e. area between the bottom and top plateau scatter bands), to the assay-inherent technical scatter (as derived from the scatter of the bottom and top plateaus, **Fig 3B** and **3C**). The formula used can be found in the Methods section.

In our hands the ADR for FluoRNT was 4.2, whereas the ADR for FRNT was 2.4 (**Fig 3D**). Since the ADR reflects expected assay fit quality in a non-linear fashion (see Methods), these ADR differences are substantial. Note that, as the top scatter is tightly distributed around 0% in both assays, the ADR is mainly dependent on the difference between the bottom scatters. We next moved to examine if these better-quality plateau parameters would result in FluoRNT

being a more reliable and predictive assay for quantifying antibody titre, when considering the full data sets to be fitted.

**FluoRNT data allows superior quality of dose-response fitting for titre determination.** Having compared the quality of the assays' plateaus for dose-response curve fitting by their ADRs, we then analysed the overall quality of the curve fitting that determines patient titre. Both assays allow fitting sigmoidal dose-response curves. However, the goodness of data fit is significantly better for FluoRNT than FRNT (median $R^2$ 0.996 vs. 0.988, p = 0.0002; Mann-Whitney test; **Fig 3E**). The actual 28 dpv titre values obtained from the fitted curves are similar for FRNT and FluoRNT, although not identical (**S3A Fig**). This similarity was expected as these assays rely on the same basic principle, but since FluoRNT provides not only better fit quality, but also better data quality, the confidence in its titres should be greater. While referencing FluoRNT and FRNT to individual pre-vaccination serum samples, instead of no-serum controls, did not greatly alter these results (**S3B and S3C Fig**). It should be noted, that the FluoRNT is more robust to different reference samples (i.e. pre-vaccination samples from 0 dpv instead of NSC controls; **S3D Fig**).

**FluoRNT data yields significantly more precise antibody titre fits than FRNT.** Having compared FluoRNT and FRNT for the analytical quality of their dose-response plateaus and their overall curve fits (**Fig 3**), we now focused on quantifying the quality of the $ED_{80}$ titre fits. For this fit, the most important region of the curve is the sloped region that brackets the $ED_{80}$ (ca. 55–90% neutralisation). Errors in this region most strongly distort the titre since these data have the most weight in $ED_{80}$ fitting. We therefore analysed the subset of data in the range of 55–90% neutralisation, calculating the absolute neutralisation difference $\partial$ between underline{technical replicates} and weighting it as a precision parameter $(1-\partial)^2$.

The median precision of FluoRNT on 28 dpv is at 94.9%, which is significantly higher than the precision of FRNT (91.7% at 28 dpv; p = 0.031, Mann-Whitney test, **Fig 3F**). This was validated again in independent assays with different virus preparations, on different patient sub-cohorts handled by different scientists (**S4 Fig**). This result shows that the precision of FluoRNT is robust to the virus stock preparation used and handling, which are the major laboratory-dependent variables. These results suggest that the greater precision of FluoRNT compared to FRNT arises from both technical aspects and the single-round-of-infection assay design.

Taken together, this shows that the data delivered by FluoRNT have good precision and are robust regardless of whether longitudinal study references or no serum references are used, or whether the YF virus stocks had been purified over a sucrose gradient or not.

## Discussion

The live-attenuated vaccine against Yellow Fever is one of the most successful vaccines ever developed. One shot of the vaccine virus YF-17D provides healthy adults with a life-long protection against the disease, and it has been successfully used for immunization for more than eighty years with a very low number of severe adverse events. YF-17D is extensively used as a model in systems biology and vaccinology studies, which require the longitudinal determination of neutralising antibody titres in a large number of samples. Recently, the need to efficiently and reliably monitor neutralising antibodies in patients or vaccinees has been highlighted. Reports of diminishing antibody titres after vaccination [24, 25], studies on the effect of booster immunizations, and the need for controlling the efficacy of vaccines that were diluted to accommodate the local need for vaccine—as happened during the 2016 outbreak of Yellow Fever in Angola and the Democratic Republic of Congo [2, 26], and the 2017–18

outbreak in Brazil [27]—are examples that underline the urgent clinical need for implementing efficient testing for neutralizing titres in large patient cohorts.

Here we describe a novel, robust, cost- and time-efficient fluorescence reduction neutralisation test (FluoRNT) utilizing a modified YF-17D virus carrying a Venus-sequence as a fluorescent reporter, to measure neutralising antibodies in human serum samples by flow cytometry. In terms of its assay workflow, FluoRNT is a labour- and time-saving alternative to existing techniques such as the widely used Focus Reduction Neutralisation Test (FRNT) and Plaque Reduction Neutralisation Test (PRNT). In terms of cost-effectiveness, FluoRNT is 10 times cheaper than the commonly used FRNT with an anti-flaviviral antibody, and even 6 times cheaper than a FRNT using the Venus reporter variant of YF-17D with an anti-GFP antibody (**S1 Table**). Most importantly however, FluoRNT is also the more precise and robust assay. While the basic diagnostic requirement in accordance with WHO standards (to determine the absence or existence of protection) is as correctly determined by FluoRNT as by the traditional FRNT, FluoRNT offers significant advantages compared to the existing assays when quantitatively determining full patient titre, both in terms of (1) assay performance, and (2) assay practicality:

(1) The primary performance criteria for assays determining neutralising titre are technical robustness and analytical quality. Through its assay design FluoRNT has inherent advantages. A major driver of robustness is its ability to measure infection after only one round of infection, by cytometry-based detection of single infected cells. Many factors other than the content of neutralising antibodies can influence infection efficiency, such as the passage number of the cells to be infected, differences in virus preparation and unspecific serum effects. Measuring reporter expression of the first round of infection restricts the impact of unspecific effects which are otherwise amplified by multiple rounds of infection. FluoRNT's cytometric detection resolves infection events even when these occur in neighbouring cells, giving its data better reliability. Due to these design features, FluoRNT data are highly reproducible within technical and biological replicates, and is well-suited for automatic dose-response curve-fitting without the requirement for manual checking or error-correction. This results in significantly more accurate and precise $ED_{80}$ determinations of protective titre in single-timepoint or longitudinal settings.

By comparison, FRNT and PRNT suffer design-dependent drawbacks in respect to precision and accuracy. They rely on multiple rounds of infection and viral release from infected cells to form detectable foci or plaques; any technical error in the assay setup will be potentiated through these rounds of infection and can add up to false-negative and false-positive results, making biologically relevant changes indistinguishable from technical scatter. FRNT and PRNT assays are more dependent on a tight characterization of the viral stock titre and ability of the target cells to form a homogeneous monolayer. The setting of these assays for new virus stocks or cells lines require more rounds of optimization compared to the FluoRNT assay. Due to variable proximity of initially infected cells, plaques or foci can merge over time (**S1 Fig** and **S1 Video**) making final measurements on plaques/foci with highly variable size and geometry difficult to quantify, both by automatic analysis tools and by eye (deviation from expected sizes or shapes, e.g. by overlap or merging). For the larger plaques in PRNT, manual determination and counting is particularly subjective and prone to experience-dependent bias (there are typically 20–100 plaques per well for the intermediate data points that influence $ED_{80}$ titres the most). For FRNT the assay readout relies on image post-processing that at least in our experience requires manual checking and interpretation for foci that are out of size specification (very small or very large) or to determine how many counts to assign to fused foci. Even for experienced experimenters it is not trivial to perform these counting/checking

steps, and they require significant time commitment and thorough blinding of personnel to give confidence in the analysis.

(2) Assays should be practicable—i.e. observable, fast, safe, and scalable. (a) An ideal assay can be evaluated at the different steps of the assay and sampled during the run-time to detect errors in the assay procedure early. While the appearance of Venus-positive cells can be followed repetitively during the FluoRNT assay via fluorescent microscopy or by sampling aliquots and analysing them in the flow cytometer, neither PRNT nor FRNT can be checked for sufficient number of foci and size before the final readout. This additional quality that brings practical value to FluoRNT is particularly valuable when working with new virus stocks, new cell batches, or samples of unknown quality. In our hands, monitoring during runs greatly simplified assay setup and sped up evaluation loops, since determining whether an assay runs "as expected" is immediately visible and can be done without interrupting or ending the experiment before the final readout. It can hence be achieved without costing significant assay manual handling or equipment time, and without reagent costs. By contrast, FRNT and PRNT require significantly more time and hands-on work to evaluate if an assay ran to completion satisfactorily or should be repeated. (b) Assays benefit from a fast turnaround and low volume needs and a low requirement of the hands-on time of skilled personnel. FluoRNT requires only <u>one round of infection</u> for detection of infection equivalent events (**S1 Video** and **S1 Fig**), which saves significant time during incubation: FluoRNT needs 24 h incubation time, compared to minimum times of 48 h for FRNT [12] and 96–120 h for PRNT [14–16]. Furthermore, FRNT and PRNT both require extensive hands-on time (e.g. layering with and removing viscous media, immunocytochemical staining, manual image checking or plaque counting).

FluoRNT's <u>stain-free readout</u> minimises total assay time by reducing the post-incubation handling steps needed before readout. This saves time of trained personnel, reduces infection risk, and avoids technical errors introduced manually during assay steps. The lower incubation time also reduces occupancy of lab equipment (incubators, fridges) needed to run the assay. Note too, that while FluoRNT can be read out as quickly as 48 h after seeding with adherent cells, it should be feasible to perform the assay on suspension cells instead, which would avoid the cell seeding/adherence/monolayer formation and terminal cell harvesting steps to further reduce the FluoRNT total assay time from setup to readout close to the minimum for first round of reporter expression.

(c) Another major advantage of FluoRNT is the objective and quantitative assay read <u>out</u> by flow cytometry. Its quantification of Venus-positive cells by a cytometer is suitable for automation and it is more reliably quantifiable than FRNT's immunostaining workup and image analysis, or the slow and subjective manual counting in the PRNT assay, both of which become time- and resource-limiting when considering assay scaleup. Particularly for population studies and for implementation in countries with strained resources, assays should be able to be run and processed in parallel, with only the least expensive assay elements (e.g. cell culture plates and media) scaling linearly to the number of assays run: which is uniquely true for FluoRNT. By contrast, for PRNT/FRNT, it is rather the most expensive elements (trained scientist time for manipulation and manual quantification, and expensive reagents such as antibodies) which scale with assay volume. FRNT's immunostaining antibody requirements add to assay cost and limits scaleup: adding the primary anti-(flavi)viral protein antibody, secondary HRP-coupled antibody and chromogenic substrate treatment is costly, requires washing after each step, is time consuming and introduces sources for handling errors.

Several considerations will inform decision-making about whether FluoRNT is a practical assay to adopt. Firstly, it requires a two-channel flow cytometer, ideally with a plate autosampler which is less likely to be available in labs performing field testing in some areas. However, for high-throughput analysis as is becoming the norm in virology, this kind of machine is

almost certainly accessible. Secondly, as for any neutralisation test, each new virus stock requires validation tests to establish and adapt the number of specific fluorescence-/focus-/plaque-forming units (FFU/PFU) for each virus preparation. Where possible, titres of large cohorts should be measured with only one virus preparation to allow better direct intra-cohort comparisons. Most assays in this study were performed with unpurified virus stocks. When we rechecked the samples with sucrose-gradient purified stocks in a second validation step, we found that FluoRNT is robust against variations in the inoculum. However, where purification via ultracentrifugation is possible, we can recommend using purified virus stocks for any neutralisation test to reduce potential variations in the virus preparation.

In order to calculate the $ED_{80}$, reference controls are needed. Here we used no-serum controls as an easily feasible control, that does not require naïve pre-vaccination samples. However, it may overestimate the antibody-dependent neutralising effect of the virus, as naïve serum exhibits some neutralising properties, which is not accounted for by no-serum controls. Therefore, we recommend the individual 0 dpv control wherever pre-vaccination controls are available. This control also contains other virus-interfering substances in the serum as well as cross-reacting antibodies and is hence a more meaningful individual control.

Finally, to adapt the FluoRNT protocol to other viruses, the generation and validation of a reporter virus is required. This is likely to be a straightforward step however, as for many viruses such constructs are already described, or can be generated by reverse genetic techniques. Depending on the original virus and the reporter virus biosafety regulations may apply.

Recognising the short-comings of FRNT and PRNT assays, other advances in in neutralisation tests for YFV have been made previously: Whiteman et al. for example assessed the FRNT's problem of overlapping foci. Instead of error prone manual counting they suggest a fluorescent readout using an imaging cytometer. However, this assay is still dependent on expensive antibody staining and the assay's progress cannot be checked during the incubation [17]. Incidentally, the same is true for earlier work of Hammarlund et al. who already developed a flow cytometry-based read-out for non-plaque forming strains of YFV, which also relies on antibody staining [28]. Matsuda et al. designed single round infectious particles for different flaviviruses. This neutralisation test is based on a luciferase system that works by the same principle as FluoRNT, using a single round of infection and directly measuring reporter expression, and can therefore be expected to be more precise than conventional neutralisation tests. However, the assay was not compared to FRNT or PRNT and the incubation time of 3 days is no improvement to FRNT or PRNT [29]. The single round of infection principle is also used by Mercier-Delarue et al. Their neutralisation test uses WNV-YF pseudotype VLPs expressing a GFP reporter that can be read out using flow cytometry. However, the test was established in a 48-well format and requires double of FluoRNT's incubation time [16].

In general, for flavivirus-serology neutralisation assays offer better specificity than assays based on the detection by ELISA of total IgM and IgG antibodies. However, cross-reactivity among flaviviruses has also been documented for neutralisation assays in areas where multiple flavivirus have recently circulated or are endemic [30]. In our YF-17D vaccination cohort assayed with FRNT and FluoRNT about half of the vaccinees have been previously vaccinated against tick-borne encephalitis (TBE) a flavivirus endemic in the south of Germany. When grouped according to TBE-vaccination status neutralisation titres of vaccinees on day 0 or day 28 measured with FRNT and FluoRNT correlated to the same extend as seen for the undivided cohort (data not shown). This argues that towards TBEV and presumably other flaviviruses FluoRNT has the same specificity as seen and described for YFV-FRNT and -PRNT-assays.

Taken together, FluoRNT has numerous advantages. It is a substantially faster, better-scalable, cheaper assay, which is more suited for high-throughput automated implementation

than the current FRNT and PRNT. It minimises handling time with infectious material as well as the time needed by trained personnel, and its progress can be estimated non-invasively using a fluorescence microscope or flow cytometer. This enables FluoRNT to measure protective titres more efficiently in terms of sample collection and processing time, as well as cohort management.

## Conclusions

FluoRNT is a robust, reproducible method with a range of analytical and practical advantages compared to current methods. We anticipate that it can significantly increase speed and accuracy of monitoring YFV exposure and provide a reliable, scalable, quick-turnaround test for protection status after vaccination with YF-17D. Clinical applications range from field testing of patients with unknown history, to sensitive longitudinal studies determining the effective dose or dilution of new and existing vaccines respectively. This method can be applied in basic research and of systems biology. Beyond research on YF-17D as a model virus, this assay can be adapted to other flaviviruses. We therefore anticipate that the FluoRNT assay could be valuably extended to other viruses, that have up to now been assayed with more traditional RNTs to determine vaccination success, as well as level and kinetics of protective titre.

## Supporting information

**S1 Note. Notes on the definition, use and inter-comparison of assay titres generated with different assay types.**
(DOCX)

**S1 Table. Overview of typical material costs per 96-well plate assay.** Costs are based on one full 96 well plate full of samples (e.g. serum of 0, 7, 14 and 28 dpv) assayed with FRNT against YF-17D or YF-Venus, or FluoRNT.
(PDF)

**S1 Video. Live cell imaging video.** Video shows Venus+ cells 28 to 100 hours after infection. Vero cells were infected with 3,000 FFU YF-17D-Venus and were overlaid with MCS after 1 hour of incubation. Frames were acquired every 30 minutes. Fluorescent cells could already be seen after 28 hours. MCS reduces the viral spread and enables the forming of foci and plaques around the primary infected cells.
(MP4)

**S1 Fig. Stills from live cell imaging video.** Stills show Venus+ cells 28, 48, 72 and 96 hours after infection with 3,000 FFU YF-17D-Venus with an MCS overlay. The flow cytometry based FluoRNT is already meaningful as early as 24 hours after infection as it does not rely on foci or plaque forming but on infected cells on single-cell level. Note that foci and plaques in close proximity to each other tend to overlap the more time passes until readout which is therefore less reliable and reproducible as the FluoRNT readout. Image processing was performed to enhance contrast.
(EPS)

**S2 Fig. Maximum infection values in different assays.** NSC values normalised to run-average NSC values. "FluoRNT", "FRNT Venus" and "FRNT 17D" display results from the main cohort of this study, whereas "FluoRNT pure 1" displays the same cohort with a purified virus. The purified virus was again tested for a second cohort ("FluoRNT pure 2" and "FRNT 17D pure"). Box and whiskers plot with 10–90 percentile.
(EPS)

**S3 Fig. Superior data quality of FluoRNT gives more robust titres regardless of the reference.** Titres obtained with FluoRNT and FRNT with NSC **(A)** or pre-vaccination samples 0 dpv **(B)** as a reference (n = 32). In both cases, FluoRNT and FRNT titres correlate significantly with each other. Spearman r. **(C)** Goodness of fit for dose-response curves for samples on 28 dpv referenced to pre-vaccination samples 0 dpv. FluoRNT gives a median $R^2$ of 0.996 vs. 0.986 for FRNT (p = 0.0001; Mann Whitney test). **(D)** Titres referenced to 0 dpv divided by titres referenced to NSC give the titre ratio. FluoRNT is slightly more robust when changing the reference (p = 0.012, Mann Whitney test). *Box and whiskers in panel C and D plot with 10–90 percentile.*
(EPS)

**S4 Fig. FluoRNT shows more precise results in different assays.** Weighted precision of FluoRNT technical replicates for 55–90% virus neutralisation is reproducibly superior to the FRNT data. Assays were performed with purified ("pure") and unpurified virus by different experimenters. All data are referenced to NSC. Box and whiskers plot with 10–90 percentile.
(EPS)

**S5 Fig. Gating strategy for the FluoRNT.** Example of flow cytometry gating strategy for Venus$^+$ cells on live cells in FluoRNT analysis.
(EPS)

**S1 File. Raw data file.**
(XLSX)

# Acknowledgments

We thank Giulia Spielmann, Natalie Röder und Christine Hörth (LMU) for assisting with study sample preparation, neutralisation assays and viral stock preparation. We thank Renate Stirner of the LMU HIV Ambulanz for excellent technical assistance and access to technical equipment. We thank Liz Schultze-Naumburg for invaluable support in building up the yellow fever vaccination cohort. We thank Charles M. Rice and Margaret MacDonald for their kind gift of the initial plasmid containing the sequence of YF-17D with the Venus-reporter sequence.

Parts of this work have been performed for the doctoral theses of MKS, LL, MZ at the Ludwig-Maximilians-Universität München.

# Author Contributions

**Conceptualization:** Magdalena K. Scheck, Simon Rothenfusser, Julia Thorn-Seshold.

**Data curation:** Magdalena K. Scheck, Lisa Lehmann, Magdalena Zaucha, Paul Schwarzlmueller, Oliver Thorn-Seshold, Julia Thorn-Seshold.

**Formal analysis:** Magdalena K. Scheck, Lisa Lehmann, Magdalena Zaucha, Oliver Thorn-Seshold, Julia Thorn-Seshold.

**Funding acquisition:** Anne B. Krug, Stefan Endres, Simon Rothenfusser, Julia Thorn-Seshold.

**Investigation:** Magdalena K. Scheck, Lisa Lehmann, Magdalena Zaucha, Paul Schwarzlmueller, Oliver Thorn-Seshold, Julia Thorn-Seshold.

**Methodology:** Magdalena K. Scheck, Giovanna Barba-Spaeth, Julia Thorn-Seshold.

**Project administration:** Simon Rothenfusser, Julia Thorn-Seshold.

**Resources:** Paul Schwarzlmueller, Kristina Huber, Michael Pritsch, Giovanna Barba-Spaeth, Anne B. Krug, Stefan Endres, Simon Rothenfusser.

**Supervision:** Giovanna Barba-Spaeth, Oliver Thorn-Seshold, Anne B. Krug, Simon Rothenfusser, Julia Thorn-Seshold.

**Validation:** Magdalena K. Scheck, Julia Thorn-Seshold.

**Visualization:** Magdalena K. Scheck, Oliver Thorn-Seshold, Julia Thorn-Seshold.

**Writing – original draft:** Magdalena K. Scheck, Oliver Thorn-Seshold, Simon Rothenfusser, Julia Thorn-Seshold.

**Writing – review & editing:** Magdalena K. Scheck, Oliver Thorn-Seshold, Simon Rothenfusser, Julia Thorn-Seshold.

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
