## [Decision Letter · Decision Letter 0]

15 Oct 2021

PONE-D-21-23143FluoRNT: A robust, efficient assay for the detection of neutralising antibodies against yellow fever virus 17DPLOS ONE

Dear Dr. Thorn-Seshold,

Thank you for submitting your manuscript to PLOS ONE. After careful consideration, we feel that it has merit but does not fully meet PLOS ONE’s publication criteria as it currently stands. Therefore, we invite you to submit a revised version of the manuscript that addresses the points raised during the review process.

Both reviewers found the article of interest but that deserved some clarification in the used vocabular and definition. Some clarifications have to be given on the comparaison with the PRNT and/or FRNT and simplification in the introduction/result/discussion that have to be done. As stated by reviewer 2 there are some other methods (with various reporter like GFP or luciferase) that have beed described to analyse seroneutralization that deserved to be discussed.

We look forward to receiving your revised manuscript.

Kind regards,

Pierre Roques, Ph.D.

Academic Editor

PLOS ONE

Journal Requirements:

2. In your Methods section, please provide additional details regarding the cell lines Vero cells and BHK-21 used in your study and ensure you have described the source. For more information regarding PLOS' policy on materials sharing and reporting, see https://journals.plos.org/plosone/s/materials-and-software-sharing#loc-sharing-materials, and for more information on PLOS ONE's guidelines for research using cell lines, see https://journals.plos.org/plosone/s/submission-guidelines#loc-cell-lines

4.  We also anticipate the potential to translate the methodology and analysis of FluoRNT to other flaviviruses such as West Nile, Dengue and Zika or to RNA viruses more generally

5. Please update your submission to use the PLOS LaTeX template. The template and more information on our requirements for LaTeX submissions can be found at http://journals.plos.org/plosone/s/latex

Reviewers' comments:

Reviewer's Responses to Questions

**Comments to the Author**

1. Is the manuscript technically sound, and do the data support the conclusions?

Reviewer #1: Yes

Reviewer #2: Yes

2. Has the statistical analysis been performed appropriately and rigorously? 

Reviewer #1: I Don't Know

Reviewer #2: Yes

3. Have the authors made all data underlying the findings in their manuscript fully available?

Reviewer #1: Yes

Reviewer #2: Yes

4. Is the manuscript presented in an intelligible fashion and written in standard English?

Reviewer #1: Yes

Reviewer #2: Yes

5. Review Comments to the Author

Reviewer #1: In the manuscript Scheck MK et al., the authors describe a neutralization assay (NT assay) for yellow fever virus (YFV) using a recombinant YFV expressing the fluorescent reporter protein Venus. The authors describe the performance of this NT assay in detail and compare the individual test steps to the protocols for plaque reduction neutralization (PRNT) and focus reduction neutralization assays (FRNT). To validate their fluorescent YFV virus-based assay (named FluoRNT), they used over 30 sera from human YFV vaccines and determined the neutralization titers resulting in 80% reduction in comparison to a focus reduction neutralization assay (FRNT). Detailed statistical analyses and dose response fitting analyses were performed to determine the accuracy and comparability of this assay. According to their data, their assay is comparable or even better with the advantage of being faster in comparison to FRNT assays. Experimental comparison to a plaque reduction neutralization assay (PRNT) was not performed, the procedures where only compared descriptive. To improve the manuscript, the authors should address the following points:

Major points:

1. Line 112 ff: for description of PRNT and FRNT – do the authors only refer to YFV assays or these assays in general? Probably only YFV – but still please specify.

2. The protocols the authors describe for PRNT and FRNT seem to be lab specific protocols (and might also depend on the cell line used) since in other publications for YFV PRNT or plaque assays are done in 3 days compared to 4-5 days as mentioned in the manuscript (line 118). Similar is probably also true for the FRNT, which is described to be done in 96 well plates which seems a rather small format, whereas the PRNT is done in 6-12 well format. Please comment and clarify better in the manuscript (since references for the described assays are missing).

3. L105: ‘The most commonly used NT assays are both reduction neutralization tests’. This sentence does not make much sense since every NT assay is a reduction neutralization assay.

I also would consider PRNT and FRNT rather similar assays (both with infectious virus) just with different read out modalities. Please adjust accordingly.

4. line 194: the authors mention that the yellow fluorescence can be seen after 3-4 days. This seems rather late. What is the titer of the YFV-Venus virus used for the studies? How does it grow in comparison to the wild-type virus?

5. line 334: the authors classify their FluoRNT assays as ‘single-round of infection assay’. This wording is confusing and should not be used since (1) virus release for YFV rather already starts at 8-12 hours post infection resulting in secondary infections until 24 h post infection. Thus ‘single-round of infection assay’ would not be correct. And (2) the term rather implies the use of real single-round infectious particles that really are only able to make one round of infection which were not used in this case.

6. line 337 ff: Description of antibody production after vaccination rather belongs to introduction than to results part.

7. Line 359ff: This whole section is only a descriptive comparison of the assays and not a real comparison with experiments.

This said – the abstract also implies that comparative analyses have been performed for both FRNT and PRNTs. However, PRNTs were not performed in comparison to FRNTs. Either the authors add them or they should make it clearer that only comparison to FRNTs was performed. The whole paragraph should be deleted in the results section and moved to the discussion, where lots of the descriptions (advantage of assay) are repeated anyway.

8. The authors only mention PRNTs and FRNTs as alternative neutralization assays. However, other NT assays have been described as well for flaviviruses like flavivirus pseudotype viruses with GFP as read out or flavivirus single round infectious particles with luciferase as read out. The authors should mention them and compare them to their assay as well in the discussion.

9. What about cross reactivity? Have the authors also tried sera against other viruses in comparison of both assays?

10. Line 591: the authors mention that the FluoRNT is robust against variations in the inoculum when using sucrose-gradient purified stocks. Does it mean that there are variations if using unpurified virus?

Minor points:

1. line 89: which ‘four continents’? Reference?

2. NT80 is more commonly used than ED80 for NT assays

3. List of costs might be relative as other antibodies/self-made antibodies might be used.

4. Is it really an anti-GFP antibody that is used for detection of Venus?

5. Line 594: adaptation to other viruses is mentioned. The authors should comment on the biosafety levels, since for some other reporter flaviviruses BSL3 laboratories might be necessary.

Reviewer #2: The authors present an FNRT method for rapid screening of neutralizing antibodies to the 17D virus. The results are clearly presented and convincing as an alternative method. FNRT has already been used and proposed by other teams for other viruses instead of PFU. Despite this last point the manuscript meets the publication criteria of PLosOne to my point (PLOS ONE does not evaluate manuscripts based on perceived significance or readership) and can therefore be accepted as is.

6. PLOS authors have the option to publish the peer review history of their article (what does this mean?). If published, this will include your full peer review and any attached files.

Reviewer #1: No

Reviewer #2: No

---

## [Author Response · Author response to Decision Letter 0]

5 Dec 2021

Reviewer #1: In the manuscript Scheck MK et al., the authors describe a neutralization assay (NT assay) for yellow fever virus (YFV) using a recombinant YFV expressing the fluorescent reporter protein Venus. The authors describe the performance of this NT assay in detail and compare the individual test steps to the protocols for plaque reduction neutralization (PRNT) and focus reduction neutralization assays (FRNT). To validate their fluorescent YFV virus-based assay (named FluoRNT), they used over 30 sera from human YFV vaccines and determined the neutralization titers resulting in 80% reduction in comparison to a focus reduction neutralization assay (FRNT). Detailed statistical analyses and dose response fitting analyses were performed to determine the accuracy and comparability of this assay. According to their data, their assay is comparable or even better with the advantage of being faster in comparison to FRNT assays. Experimental comparison to a plaque reduction neutralization assay (PRNT) was not performed, the procedures where only compared descriptive. To improve the manuscript, the authors should address the following points:

Major points:

1. Line 112 ff: for description of PRNT and FRNT – do the authors only refer to YFV assays or these assays in general? Probably only YFV – but still please specify.

We refer primarily to YFV and made that clearer in this section of the manuscript. But of course, the same principles and considerations apply similarly to other viruses as stated in lines 113 and 129 ff. PRNT and FRNT described here for YFV work the same way for NTs of any other viruses that form plaques (enabling PRNT) or for which antibodies exist to detect viral infection (enabling FRNT).

2. The protocols the authors describe for PRNT and FRNT seem to be lab specific protocols (and might also depend on the cell line used) since in other publications for YFV PRNT or plaque assays are done in 3 days compared to 4-5 days as mentioned in the manuscript (line 118). Similar is probably also true for the FRNT, which is described to be done in 96 well plates which seems a rather small format, whereas the PRNT is done in 6-12 well format. Please comment and clarify better in the manuscript (since references for the described assays are missing).

Yes, the assay protocols are described as we use them in our lab. Published YFV FRNT/PRNT protocols vary slightly from lab to lab (e.g. both shorter and longer timelines depending on the cell line and the virus stock used); however, the time lines are representative of typical procedures and we have added a more general commentary on this (lines 98ff). Thank you for pointing out the missing references; we have added references for these timelines (lines 120, 131 and 135).

3. L105: ‘The most commonly used NT assays are both reduction neutralization tests’. This sentence does not make much sense since every NT assay is a reduction neutralization assay.

I also would consider PRNT and FRNT rather similar assays (both with infectious virus) just with different read out modalities. Please adjust accordingly.

Yes, thank you for catching it, we have fixed it (line 92 f.); and yes, PRNT and FRNT are indeed similar from the setup, as we say (line 129). We describe them sequentially since (a) they are the most widely used neutralisation tests, so in this way we address the target readership no matter which assay they use; and (b) showing the advances in our paper relies on understanding the influence of readout modality on the precision, accuracy, and practicality of assays: so, a more detailed description of the two assay readouts to us seems adequate.

4. line 194: the authors mention that the yellow fluorescence can be seen after 3-4 days. This seems rather late. What is the titer of the YFV-Venus virus used for the studies? How does it grow in comparison to the wild-type virus?

Thank you for the opportunity to clarify this. After infection with YF-Venus, yellow fluorescence is seen within a day, as we state in e.g. line 576 ff. and illustrated in Supporting Video 1. This should match the reader’s and reviewers’ expectations. Line 194 (now line 191) mentioning “3-4 days” instead referred to electroporating cells with YF-Venus-RNA for YF-Venus virus production, which cannot be compared to how the vaccine strain grows. We edited line 191 to make this clear to the reader.

 Please note that each assay type uses different measures of infectivity (e.g. FRNT => FFU, PRNT => PFU), as they operate on different time scales and have different readouts into which the functional infective units have to translate. The relevant descriptor for the virus stock in FluoRNT is the percentage of Venus+ cells obtained 24 hours post inoculation. In our protocol we aim for approximately 25% Venus+ cells (line 269); to achieve that we typically had to use approximately 5000 PFU or 2875 FFU/well/per 25.000 Vero cells of our YF-17D-Venus Stocks. That makes the amount of virus used in the FluoRNT 22-fold more concentrated than in the FRNT (line 270 ff.). As described in the section on virus stock production (lines 195 ff.) the YF-17D-Venus stock used for this study propagates somewhat slower than the YF-17D stock derived from the Stamaril vaccine that was used for this study.

5. line 334: the authors classify their FluoRNT assays as ‘single-round of infection assay’. This wording is confusing and should not be used since (1) virus release for YFV rather already starts at 8-12 hours post infection resulting in secondary infections until 24 h post infection. Thus ‘single-round of infection assay’ would not be correct. And (2) the term rather implies the use of real single-round infectious particles that really are only able to make one round of infection which were not used in this case.

Thank you for the opportunity to clarify what for our paper is a very important point. We have modified the manuscript to define “single round of infection” more precisely (lines 147, 336, 533 ff. and 576 ff.). FluoRNT readout is reachable when the first round of cells has become infected and express sufficient Venus reporter for measurement (see Fig S1 and Video S1). Regarding part (1): Video S1 particularly highlights the time-separation between the waves of reporter expression in the first and second rounds of infection. Given our timeline of measurement at 24 h and the somewhat slower kinetics of the YF-17D-Venus stock compared to the YF-17D stock, we are confident that we are indeed only detecting first-round infected cells. Regarding part (2): yes, technically YF-17D-Venus can progress to multiple rounds of infection, but since conceptually the FluoRNT assay neither requires subsequent rounds of infection, nor runs long enough for them to happen, we agree that “first” is a better description than “single”, implying there could be second, third, fourth rounds of infection, but they are not needed for the assay readout.

6. line 337 ff: Description of antibody production after vaccination rather belongs to introduction than to results part.

Thank you for the suggestion. However, we do feel it is important to mention this in the results part. Understanding the antibody production time course is essential to interpret results from the different post vaccination days. We would therefore rather leave it in the results section than making it a part of the general introduction to NTs and motivation for the FluoRNT assay (now line 339 ff.).

7. Line 359ff: This whole section is only a descriptive comparison of the assays and not a real comparison with experiments. This said – the abstract also implies that comparative analyses have been performed for both FRNT and PRNTs. However, PRNTs were not performed in comparison to FRNTs. Either the authors add them or they should make it clearer that only comparison to FRNTs was performed. The whole paragraph should be deleted in the results section and moved to the discussion, where lots of the descriptions (advantage of assay) are repeated anyway.

Thank you for the constructive suggestions. (1) We have streamlined this section (now line 366 ff.) and incorporated most of it in the discussion part. (2) We have made it clear now, that our direct experimental comparisons were only done between FRNT and FluoRNT, and comparisons with PRNT are purely descriptive (lines 370 ff.). That being said, as the reviewer also stated (item 3), FRNT and PRNT are in many ways such similar assays that we feel this experimental comparison is sufficient to highlight the benefits of FluoRNT compared to both assays: and all the more so, since FRNT is in nearly all practical respects the “better” assay than PRNT (faster, better data quality, less by-eye counting, etc.), so we feel our comparisons to FRNT are the more stringent ones that could be chosen.

8. The authors only mention PRNTs and FRNTs as alternative neutralization assays. However, other NT assays have been described as well for flaviviruses like flavivirus pseudotype viruses with GFP as read out or flavivirus single round infectious particles with luciferase as read out. The authors should mention them and compare them to their assay as well in the discussion.

Correct, we agree and implemented comparisons with other neutralisation assays as suggested (e.g. cited other advances on the PRNT- and FRNT-type assays, including the Luciferase reporter assay published by Matsuda et al. and the pseudotype reporter assay by Mercier-Delarue et al.; lines 631 ff.). We feel that the current state of the manuscript is now fair, by referencing these additional methods appropriately, yet keeping the reader’s focus and reserving the most discussion space for the most widely used neutralisation tests (with which we have previously worked).

9. What about cross reactivity? Have the authors also tried sera against other viruses in comparison of both assays?

A very good point, thank you. Our patient cohort is naïve for all flaviviruses except for TBEV. TBEV is endemic in southern Germany and about half of the vaccinees had been previously vaccinated against TBE. Anti-YFV titres of previously TBEV vaccinated participants do not differ depending on whether they were measured by FRNT or FluoRNT, suggesting, that FluoRNT is as robust to cross-reactivity as FRNT is. Data how previous vaccination influences the immune response to the YF-17D vaccine forms part of a larger study on viral cross-reactivity that will be published elsewhere and is therefore not given in this manuscript. However, in lines 647 ff. we have now included a statement to cross-reactivity and describe this equivalent robustness towards TBEV of FRNT and FluoRNT.

10. Line 591: the authors mention that the FluoRNT is robust against variations in the inoculum when using sucrose-gradient purified stocks. Does it mean that there are variations if using unpurified virus?

An important question. From day to day lab work we know that there are stock-dependent variations for both types of stocks purified as well as unpurified However, conceptually, it is of course easier to claim and show robustness between purified stocks where most contaminants are eliminated, and possibilities for variations are restricted. So, in general, we expect that any assay can benefit from using purified virus if possible. We have made it clearer in the manuscript now, that this is not a FluoRNT-specific recommendation (line 616 ff.). When stock-dependent data for FluoRNT are plotted on the same scale as the same stock-dependent data acquired using FRNT, our conclusions regarding the assay’s precision and data quality stay true regardless whether unpurified or purified virus stocks were used (as shown in Supporting Figures S2 and S4). Therefore, we feel confident to claim that the FluoRNT is essentially robust to the use of unpurified virus, at least in our hands.

Minor points:

1. line 89: which ‘four continents’? Reference? 

Yes, this should have been just Africa & South America, changed and referenced (now line 75).

2. NT80 is more commonly used than ED80 for NT assays

We have clarified additionally in line 91, that we write ED for what is also called EC or NT.

3. List of costs might be relative as other antibodies/self-made antibodies might be used.

True; though we find it more helpful to give a typical estimate rather than not to give one, and we consider our estimation quite reasonable; given that cost is just one of the factors we discuss that need to be weighed when choosing an assay, we would like to keep it as it is.

4. Is it really an anti-GFP antibody that is used for detection of Venus?

Yes, since Venus is a variant of GFP, the 4-9 mutations do not affect the antibody binding site.

5. Line 594: adaptation to other viruses is mentioned. The authors should comment on the biosafety levels, since for some other reporter flaviviruses BSL3 laboratories might be necessary.

Yes, thank you, we noted this in the manuscript line 629.

In conclusion, we thank Reviewer 1 for her/his detailed reading and suggestions which we have addressed with the described manuscript changes.

Reviewer #2: The authors present an FNRT method for rapid screening of neutralizing antibodies to the 17D virus. The results are clearly presented and convincing as an alternative method. FNRT has already been used and proposed by other teams for other viruses instead of PFU. Despite this last point the manuscript meets the publication criteria of PLosOne to my point (PLOS ONE does not evaluate manuscripts based on perceived significance or readership) and can therefore be accepted as is.

We thank the reviewer for this positive evaluation.

---

## [Decision Letter · Decision Letter 1]

17 Dec 2021

FluoRNT: A robust, efficient assay for the detection of neutralising antibodies against yellow fever virus 17D

PONE-D-21-23143R1

Dear Dr. Thorn-Seshold,

We’re pleased to inform you that your manuscript has been judged scientifically suitable for publication and will be formally accepted for publication once it meets all outstanding technical requirements.

Kind regards,

Pierre Roques, Ph.D.

Academic Editor

PLOS ONE

Additional Editor Comments (optional):

Reviewers' comments:

Reviewer's Responses to Questions

**Comments to the Author**

1. If the authors have adequately addressed your comments raised in a previous round of review and you feel that this manuscript is now acceptable for publication, you may indicate that here to bypass the “Comments to the Author” section, enter your conflict of interest statement in the “Confidential to Editor” section, and submit your "Accept" recommendation.

Reviewer #1: All comments have been addressed

2. Is the manuscript technically sound, and do the data support the conclusions?

Reviewer #1: Yes

3. Has the statistical analysis been performed appropriately and rigorously? 

Reviewer #1: Yes

4. Have the authors made all data underlying the findings in their manuscript fully available?

Reviewer #1: Yes

5. Is the manuscript presented in an intelligible fashion and written in standard English?

Reviewer #1: Yes

6. Review Comments to the Author

Reviewer #1: The authors addressed the comments very nicely and the manuscript can now be accepted as it is for publication.

7. PLOS authors have the option to publish the peer review history of their article (what does this mean?). If published, this will include your full peer review and any attached files.

Reviewer #1: No

---

## [Editor Report · Acceptance letter]

24 Jan 2022

PONE-D-21-23143R1 

FluoRNT: A robust, efficient assay for the detection of neutralising antibodies against yellow fever virus 17D 

Dear Dr. Thorn-Seshold:

I'm pleased to inform you that your manuscript has been deemed suitable for publication in PLOS ONE. Congratulations! Your manuscript is now with our production department. 

Kind regards, 

on behalf of

Dr. Pierre Roques 

Academic Editor

PLOS ONE